# Physiological and Metabolic Responses of Rice to Reduced Soil Moisture: Relationship of Water Stress Tolerance and Grain Production

**DOI:** 10.3390/ijms20081846

**Published:** 2019-04-15

**Authors:** Jinyoung Y. Barnaby, Jai S. Rohila, Chris G. Henry, Richard C. Sicher, Vagimalla R. Reddy, Anna M. McClung

**Affiliations:** 1USDA-ARS Dale Bumpers National Rice Research Center, Stuttgart, AR 72160, USA; jinyoung.barnaby@ars.usda.gov (J.Y.B.); jai.rohila@ars.usda.gov (J.S.R.); 2Department of Biological and Agricultural Engineering, University of Arkansas, Fayetteville, AR 72701, USA; cghenry@uark.edu; 3USDA-ARS Adaptive Cropping Systems Laboratory, Henry A. Wallace Beltsville Agricultural Research Center, Beltsville, MD 20705, USA; rsicher1981@gmail.com (R.C.S.); vangimalla.reddy@ars.usda.gov (V.R.R.)

**Keywords:** rice, drip irrigation, metabolite profile, photosynthetic adjustment, yield response, stress adaptation

## Abstract

Access to adequate irrigation resources is critical for sustained agricultural production, and rice, a staple cereal grain for half of the world population, is one of the biggest users of irrigation. To reduce water use, several water saving irrigation systems have been developed for rice production, but a reliable system to evaluate cultivars for water stress tolerance is still lacking. Here, seven rice cultivars that have diverse yield potential under water stress were evaluated in a field study using four continuous irrigation regimes varying from saturation to wilting point. To understand the relationship between water stress and yield potential, the physiological and leaf metabolic responses were investigated at the critical transition between vegetative and reproductive growth stages. Twenty-nine metabolite markers including carbohydrates, amino acids and organic acids were found to significantly differ among the seven cultivars in response to increasing water stress levels with amino acids increasing but organic acids and carbohydrates showing mixed responses. Overall, our data suggest that, in response to increasing water stress, rice cultivars that do not show a significant yield loss accumulate carbohydrates (fructose, glucose, and myo-inositol), and this is associated with a moderate reduction in stomatal conductance (*g_s_*), particularly under milder stress conditions. In contrast, cultivars that had significant yield loss due to water stress had the greatest reduction in *g_s_*, relatively lower accumulation of carbohydrates, and relatively high increases in relative chlorophyll content (SPAD) and leaf temperature (Tm). These data demonstrate the existence of genetic variation in yield under different water stress levels which results from a suite of physiological and biochemical responses to water stress. Our study, therefore, suggests that in rice there are different physiological and metabolic strategies that result in tolerance to water stress that should be considered in developing new cultivars for deficit irrigation production systems that use less water.

## 1. Introduction

Rice is recognized as a primary food source for more than 50% of the global population [1] and the USA is among the major world exporters of rice [2] with Arkansas producing half of the country’s crop. Like in many parts of the world, Arkansas rice production, under typical paddy management practices, is not sustainable due to current rates of ground water depletion, the primary source for crop irrigation [3]. However, several management options are available that conserve water in rice production, via deficit irrigation. These options include precision leveled fields with straight levees, zero grade fields, pivot irrigation, furrow irrigation, multiple inlet irrigation, and intermittent or alternate wetting and drying (AWD) irrigation [4]. Although these options can conserve water, such practices will not be adopted if crop yield is reduced. It is essential for the future sustainability of rice production to identify and develop rice cultivars that can maintain or increase yields under deficit irrigation [5]. Consequently, having an accurate means of evaluating breeding lines for yield potential and stress response under water deficits is necessary for identification and development of new rice cultivars that will help to preserve limited water resources.

Traditionally, rice is grown in flooded paddies resulting in anaerobic soil conditions. Less is known about the impact of irrigation systems where the soil is not saturated such as alternating wetting and drying, furrow, sprinkler, and subsurface drip irrigation (SDI) systems. SDI has been used to efficiently irrigate high value horticulture crops, agronomic crops, and turf grass [6] although it is not commercially viable for rice production [7]. However, this system can be used to deliver distinct irrigation levels as a means to evaluate stress responses among rice cultivars in order to minimize potential yield penalties under aerobic irrigation systems which is a major concerns for rice growers. Two critical plant stages that are sensitive to water stress conditions are the transition phase from vegetative to reproductive stage and the grain fill stage [8]. Water stress during the transition stage results in a reduction in photosynthetic activity and osmotic imbalances in plant leaves [9,10] and this, like stress during the grain fill stage, can result in a reduction in number and size of marketable grain. Rice farmers avoid any water stress from the time of flowering (heading) through the early grainfill stages because of the risk of significant impacts on yield and quality. Therefore, evaluating stress responses to identify tolerant cultivars during the transition stage is important for preserving rice yields.

Since grain yield is a function of all the molecular, biochemical, and physiological processes occurring in the plant throughout the cropping season, a major focus of current worldwide research has shifted towards -omics [11] where the relative production of transcripts [12,13], proteins [14], and/or metabolites [15] can be observed. Metabolomics is one such technology that has the capability to enhance crop breeding efforts because metabolites are directly related to the phenotype as they reveal the products of the genome and protein outputs [16,17,18]. Metabolomics has been used to identify drought biomarkers in a range of rice cultivars grown under flooded, rain-fed lowland, and upland conditions [15].

Due to the timing and duration of when drought may occur, a wide variety of stress responsive traits may be impacted and thus, breeding for drought tolerance in rice is considered complexly inherited [19]. However, depending on the genetic materials used and method of inducing water stress, major QTLs for drought tolerance have been identified [20]. Moreover, transgenic technology has been used to demonstrate the role of specific genes and metabolic pathways that impact tolerance to drought stress [21,22,23]. Although these methodologies are helping to elucidate the genetic control of drought tolerant processes in rice, improvement of rice cultivars for production in the USA largely rely upon conventional or marker assisted breeding approaches since, currently, transgenic varieties are not allowed. To this end, we evaluated 15 cultivars/germplasm accessions, well adapted for production in the southern USA, that could be used in future breeding programs and selected seven cultivars displaying varied yield response to water stress for our study.

Here, we sought to determine whether physiological and metabolic characteristics could be used to evaluate yield variation in rice grown under deficit irrigation. Seven cultivars previously evaluated for yield response to water stress were grown under field conditions using an SDI system that applied four separate irrigation treatments throughout the season. The physiological and metabolic assessment was performed just prior to heading (panicle initiation stage) as the plant transitioned from the vegetative to the reproductive stage. The objectives of the study were to use an SDI system to deliver a range in irrigation regimes (1) to quantify and characterize changes in the physiological and metabolic responses of seven rice cultivars varying in yield reduction to reduced soil water availability, and (2) to assess whether these responses were associated with final grain yield, in order to identify traits that can be selected in breeding for new cultivars that have greater water use efficiency and higher yield under deficit irrigation.

## 2. Results

### 2.1. Rice Cultivar Selection for Differences in Yield Response to Reduced Soil Water Availability

Seven cultivars were chosen based on the differences in yield response across the four irrigation levels (Table 1 and Appendix A) from the 2014 and 2015 data, and were analyzed for physiological and metabolic parameters in the 2016 field experiment. For each year, the cultivar (C) and irrigation (I) effects were significantly different (*p <* 0.01) whereas their interaction was not (Appendix A). Based on their yield performance over the three years (Table 1), the cultivars were divided into three categories, i.e., high, intermediate, and low response to water stress. The high response group included Teqing (TQNG) and PI312777 (PI77) (*p* < 0.01; 1.6 < average slope < 2.4) and the low or non-response group included Francis (FRCS), Kaybonnet (KBNT), and Saber (SABR) (ns), whereas Lagrue (LGRU) and Lemont (LMNT) were intermediate in their responses (0.05 < *p* < 0.01; 0.8 < average slope < 1.3) (Table 1). Although TQNG and PI77 had the greatest response to water deficits, they also had the highest yield under IRRI_1. The ranking of the cultivar yield differences and slopes were similar in each of the three years although the response to water deficits were not significant in 2015, except for one cultivar (Table 1). Overall air and soil temperatures as well as total solar fluxes during both the vegetative/reproductive stages (from planting to heading) and grain fill stage (from heading to harvest) in 2015 were lower compared to those in 2014 and 2016 suggesting yield potentials may have been limited due to weather conditions in 2015 (Appendix A).

### 2.2. Irrigation Treatments Resulted in Distinct Differences in Physiological Responses

Seasonal amounts of water applied for the four irrigation levels in 2016 were 87, 76, 63, and 44 ha·cm/ha at IRRI_1 to 4, respectively (Appendix A). To determine the soil volumetric water content (VWC) for each irrigation treatment, means and standard errors were determined using the soil VWC measured on each experimental unit throughout the season, prior to any rainfall event, including at 84 days after emergence (DAE) when samples were collected for metabolite analysis. Average measured volumetric soil water contents were 36% VWC, 28% VWC, 20% VWC, and 16% VWC at IRRI_1, 2, 3, and 4, respectively (Figure 1A). Significant plant stress responses were observed for four physiological traits including relative chlorophyll content (SPAD) index, leaf temperature (Tm), net CO_2_ assimilation (A), and stomatal conductance (*g_s_*) (Figure 1B–E). Overall SPAD index, an indirect measurement of chlorophyll content, increased up to IRRI 3, but then no difference between IRRI 3 and IRRI_4 was observed (Figure 1B). With increasing water stress, leaf temperature increased (Figure 1C), photosynthesis decreased (Figure 1D), and stomatal conductance decreased (Figure 1E).

### 2.3. Yield Potential of Cultivars under Non-Stressed Conditions Is Not Predictive of Yields under Severe Water Stress 

Averaged over the three years, both high response cultivars produced high yields (> 38 g/plant) under the non-water stressed conditions and the percent yield loss at IRRI_4 as compared to IRRI_1 was also the highest (> 60%) (Table 2). The three-year average yield potential under IRRI_1 for the intermediate and low response cultivars ranged from 14.43 g/plant for KBNT to 35.61 for FRCS and the percent yield loss at IRRI_4 ranged from 18.35% (SABR) to 55.33% (LMNT). Although FRCS and SABR had lower yields at IRRI_1 than TQNG and PI77, they had the highest average yield under the most water stressed condition (IRRI_4), and thus are designated as low response cultivars. Although there was no significant cultivar by irrigation (C x I) interaction for yield in each individual year (Appendix A) when pooled across years, the cultivars with the highest yield under non-stressed conditions, did not have the highest yield under the most stressed irrigation regime. In 2016, the high and intermediate response groups had the most dramatic percent yield loss at IRRI_4 as compared to the low response cultivars (Table 2). The percent yield loss of the low response cultivars, KBNT, SABR, and FRCS, was less than 22% under the severe stress condition (IRRI_4) compared to that under the water saturated condition (IRRI_1) in 2016. Among intermediate or high response cultivars in 2016, the yield reduction was more than 39% under the severe stress condition with LMNT showing the highest reduction (76.7%) (Table 2).

### 2.4. Association of Physiological Changes among Cultivars That Differ in Yield Response to Water Stress 

All seven cultivars responded to reduced irrigation levels in that A and *g_s_* decreased, and relative chlorophyll content (SPAD index) and leaf Tm increased (Appendix A). Per genotypic variation, PI77 displayed the greatest water stress-induced decrease in *g_s_* and increases in SPAD index while TQNG showed the greatest decrease in A and increase in leaf Tm and was second only to PI77 in significant responses in *g_s_* and SPAD index (*p* < 0.01, Appendix A). Relative to the other cultivars, LGRU showed moderate decrease in A and *g_s_*, and a moderate increase in Leaf Tm (*p* < 0.01) but no significant response in water use efficiency (WUE) and SPAD index (ns) (Appendix A). LMNT showed relatively greater increase in WUE in response to increasing water stress (*p* < 0.01, Appendix A) while FRCS and KBNT had greater decrease in A (*p* < 0.01, Appendix A). Furthermore, KBNT showed the greatest decrease in WUE (*p* < 0.05, Appendix A). Lastly, SABR showed a significantly greater decrease in leaf Tm (*p* < 0.01, Appendix A).

Two-way hierarchical clustering (HC) analysis (distance: Euclidean distance) was performed to evaluate cultivar differences in yield and physiological traits (A, *g_s_*, photosynthetic WUE (A/*g_s_*), SPAD, and Leaf Tm), which showed statistical significance for C, I, and/or C x I effects (0.05 < *p* < 0.01, Appendix A) (Figure 2). While bivariate analysis only looks at one trait as a function of increasing water stress levels, HC analysis presents a holistic view of how all observed traits of the seven cultivars are related to water stress. In our study, HC analysis showed two contrasting patterns, decreases in A, *g_s_* and yield traits and increases in WUE, SPAD index and leaf Tm, in response to increasing water stress levels (from IRRI_1 to IRRI_4) (Figure 2, x-axis), and this contrasting pattern generated three groups (Figure 2, y-axis). Group I represented water saturated and mild stress conditions (IRRI_1 and IRRI_2) displaying low SPAD and leaf Tm. Group II included a mixture of responses but identified all cultivars under the moderate stress condition (IRRI_3) with these generally having an increase in SPAD and leaf Tm. The Group III cluster was based on a similar response pattern, i.e., having lower in yield, A and *g_s_*, but higher in SPAD, WUE, and leaf Tm. We also performed principal component analysis (PCA) and found, similar to the HC analysis, that PC1 accounted for 50.6% variation and separated IRRI_4 from IRRI_1 to 3 due to the contrasting pattern of A, *g_s_*, and yield from SPAD, leaf Tm and WUE (Appendix A).

There were several distinctions (Figure 2 and Appendix A): (1) Among the three highest yielding cultivars in 2016 under IRRI_1 (Table 1), FRCS had only 22% yield reduction even under severe stress (IRRI 4) (thus, categorized as a low response cultivar) (Table 2) and generally displayed higher levels in SPAD index but lower levels in *g_s_* under all water deficit conditions compared to the rest of the two high yielding and high yield response cultivars, TQNG and PI77; (2) unlike the other low yield cultivars (KBNT and SABR), LGRU had relatively elevated SPAD index among cultivars at IRRI_1 and this did not change with increasing water stress levels; and (3) WUE significantly increased in all cultivars except SABR and FRCS under the most severe stress condition (IRRI_4; Group III) in our study.

### 2.5. Metabolic Responses of Seven Cultivars under Reduced Irrigation Levels Resulted in Four Distinctive Cluster Groups

To characterize the metabolic changes in response to reduced irrigation levels, primary foliar metabolites, which are directly involved in growth, development and reproduction, were examined using the seven cultivars in 2016. Out of 40 metabolites measured, 29 showed a significant C, I, and/or C x I effect (*p <* 0.05), and were selected for further analysis (Appendix A).

Like the physiological traits above, metabolites were also divided largely based on water stress levels. In response to increasing water stress, four clusters of metabolites (Figure 3, y-axis) and two groups (Figure 3, x-axis) among the seven rice cultivars were identified. The yellow indicated increased levels at each IRRI_2, 3, and 4 as compared to IRRI_1, and the blue represented decreased values at each IRRI treatment compared to 0% water deficit condition. Cluster 1 included mostly carbohydrates, i.e., fructose, glucose, myo-inositol, and trehalose. In general, Cluster 1 metabolites accumulated under milder water stress (IRRI_2 and 3) but were reduced under severe water stress (IRRI_4) as compared to the water saturated condition (IRRI_1). Distinctively, two high yielding cultivars, PI77 and TQNG, did not accumulate Cluster 1 metabolites as much as other cultivars did at IRRI_2, but this pattern was the opposite at IRRI_4, i.e., higher accumulation than other cultivars. Cluster 2 metabolites consisted mostly of organic acids, which decreased under all water stress conditions (IRRI_2 to 4). Conversely Cluster 3, a mix of organic acids, amino acids, and carbohydrates, showed diverse response patterns across cultivars and irrigation treatments. Cluster 4 contained mostly amino acids, and their concentrations increased with water stress, with the greatest response observed at IRRI_4. Based on the pattern of Cluster 1 to 4 metabolites, cultivar xirrigation effects were divided into two groups (Figure 3, x-axis). Group 1 contained the cultivars mostly under the mild (IRRI_2) and moderate stress (IRRI_3) levels, while Group 2 included the ones mostly under the severe stress (IRRI_4) and some at the moderate stress (IRRI_3) level. Interestingly, for PI77, a high yielding cultivar having with high yield reduction in response to increasing water stress, all irrigation responses (highlighted in green letters, Figure 3) were in Group 1, indicating that, regardless of the severity of the irrigation treatment, PI77 responds in a consistent manner relative to IRRI_1. In contrast to PI77, all irrigation responses of SABR, a low yielding cultivar having with low yield reduction with increasing water stress, (highlighted in orange letters, Figure 3) were in Group 2 along with cultivars that were under more severe water stress (IRRI_3 and 4) implying that even at mild water stress conditions (IRRI_2), SABR responds similar to other cultivars under more severe irrigation stress. TQNG was intermediate between PI77 and SABR, with responses that clustered in both Groups 1 and 2. PCA analysis also showed that the major variation (PC1) was due to water stress levels, IRRI_1, 2, 3, to 4 (27.5% difference), and the second largest variation was due to separation of IRRI_2 and 3 from IRRI_1 and 4 (11% difference), and this was largely due to the Cluster 1 metabolites, Frc, Glc, myo-ino, and Tre (Appendix A).

### 2.6. High vs. Non- or Low Yield Water Stress Response Cultivars Displayed Distinctive Metabolic Patterns

When all treatments (cultivars and irrigations) are combined for clustering analysis, intraspecific differences in metabolic patterns can be obscured. We, therefore, ran a hierarchical clustering for each cultivar and two distinctive metabolic patterns were observed (Group 1 and 2, Figure 4). First, Cluster 1 (blue letters) (fructose, glucose, myo-inositol, and trehalose - known as osmo-protectants) and Cluster 4 compounds (red letters) (branched amino acids, proline, raffinose, mannitol, and putrescine), as designated in Figure 3, remained essentially distinct in Groups 1 and 2, respectively, for LGRU, an intermediate yield response cultivar, and FRCS, KBNT, and SABR, all low yield response cultivars (Figure 4D–G). However, these metabolites were primarily clustered together in Group 2 in the two high yield response cultivars (TQNG and PI77) and in LMNT (Figure 4A–C). Furthermore, for the intermediate and non- or low response cultivars (LGRU, FRCS, *SABR*, and KBNT), Cluster 1 metabolites (blue letters) were significantly increased at the mild and moderate stress condition (IRRI_2 and IRRI_3), but they were unchanged or decreased under the severe water stress condition (IRRI_4) (Figure 4D–G). In contrast, Cluster 1 metabolites tended to also increase at IRRI_4 in the high response cultivars (PI77 and TQNG). Generally, Cluster 4 metabolites (red letters) had greater accumulation as water stress levels increased in all seven cultivars, but this response was more muted for FRCS.

### 2.7. Cultivars with Little Loss in Yield Due to Reduced Irrigation Levels May have Ability to Regulate Osmo-Protectants

To determine the relationships between yield and physiological traits in response to water stress, correlation coefficients were calculated using data from all four irrigation levels (IRRI_1 to IRRI_4). Yield was most strongly and positively correlated with *gs* (0.53, *p* < 0.01) and A (0.43, *p* < 0.01) (Appendix A). To further evaluate cultivar differences due to water stress, yield and physiological traits were correlated with the contents of 29 metabolites using two-way hierarchical clustering analysis (distance: Pearson correlation) and PCA (Figure 5A–G, Appendix A). In general, the Cluster 4 metabolites (amino acid group, red letters) showed positive correlations with leaf Tm, SPAD, and WUE and negative correlations with A, *gs* and yield across all cultivars except for KBNT where Cluster 4 metabolites were also positively correlated with yield. This suggests that Cluster 4 metabolites are stress responsive metabolites. Similarly, Cluster 1 metabolites (carbohydrate group, blue letters) were observed to have positive correlations with WUE, SPAD, and/or leaf Tm traits and negative correlations with A, *gs* and yield but this occurred only in the cultivars with the most yield reduction in response to stress (TQNG, PI77, and LMNT) (Figure 5A,B,D, Table 2). The opposite was observed for Cluster 1 metabolites in cultivars with the least or no reduction in yield (LGRU, SABR and KBNT) (Figure 5E–G, Table 2). Unlike the other low yield response cultivars, Cluster 1 metabolites in FRCS, a low response cultivar with high yield, respond in a pattern similar to that of the two high yielding cultivars, TQNG and PI77 (Figure 5A–C).

Visible wilted and leaf curling symptoms were observed in response to severe stress (IRRI_4) in the intermediate and no or low yield response cultivars (LMNT, LGRU, SABR, and KBNT) (Appendix A). Mild and moderated stress (IRRI_2 and IRRI_3) did not result in any visible stress symptoms in any of cultivars. Furthermore, the three highest yielding cultivars (TQNG, PI77, and FRCS) did not display any wilting symptoms even under severe stress (IRRI_4) (Appendix A). Wilted and leaf curling symptoms were coincident with negative correlation between leaf Tm and Cluster 1 (blue letters) compounds (Figure 5D–G).

To illustrate the stress response of each trait in relation to yield, two high yield response (TQNG, PI77), two intermediate (LGRU, LMNT), and three low yield response (FRCS, SABR, KBNT) cultivars were combined respectively, and plotted as a function of increasing water stress, i.e., reduced volumetric soil water content as a result of the four irrigation treatments. The physiological (Appendix A) and metabolic (Appendix A) traits that showed a consistent significant response to reduced soil water availability among the three yield response groups (probability > F, *p* < 0.0001) were chosen here (Appendix A). Given the large number of metabolites that had a significant response, six that were representative were chosen for presentation. High response cultivars had the greatest reduction in yield and *g_s_* (Figure 6A,C) while changes in A were largely due to irrigation treatments with less differences seen among the three yield response groups (Figure 6B). Among Cluster 1 metabolites, fructose, glucose, and myo-inositol, contents varied significantly in response to irrigation treatments for each of the yield response groups (*p* < 0.001) (Appendix A). These metabolites accumulated the most at the mild and moderate stress levels (28 and 20% VWC; IRRI_2 and 3) in the intermediate and low response cultivars, while this peak appeared at the moderate stress level (20% VWC; IRRI_3) in the high response cultivars (Figure 6D–F). These results demonstrated a capacity of lower yield response cultivars to increase leaf soluble sugars at milder stress level compared to high yield response cultivars. For Cluster 4 metabolites, all increased with increasing water stress regardless of cultivars (Figure 6G–I).

## 3. Discussion

### 3.1. Water Management Practice

The majority of U.S. rice is grown in the mid-south and is irrigated using the Mississippi River Valley Alluvial (MRVA) aquifer. As overdraft of the MRVA aquifer has increased, there is an incentive to improve irrigation efficiency [4]. For rice production, efforts are underway to evaluate whether alternate wetting and drying (AWD) can serve as a means to reduce irrigation inputs while maintaining yields. Using an SDI system, we implemented four irrigation treatments that ranged in soil moisture levels from field capacity (30% VWC) to permanent wilting point (14%VWC) from which the response of 15 genotypes to water stress could be evaluated.

### 3.2. Diverse Yield Responses to Water Stress Observed among Different Cultivars

Based on the initial evaluation in 2014 and 2015, seven cultivars that significantly differed for yield response to soil water deficits were selected for a detailed evaluation of physiological and metabolic traits in the third year (2016). When averaged over three years, yield reduction due to increasing water stress was greatest in high yielding cultivars, TQNG and PI77, whereas LGRU and LMNT had intermediate responses, and yield responses of FRCS, KBNT, and SABR were relatively stable. Our results demonstrate that yield response to water stress differs among these cultivars, although they were developed for production using non-stressed, season long flooded fields. Among this set of cultivars, TQNG and PI77 effectively utilized water resources under water-saturated conditions and produced high yield, but when stressed, they were not resilient and demonstrated the greatest losses in yield. In contrast, the other cultivars differed in their response to water stress and ability to maintain yield under fully irrigated conditions. These different response mechanisms demonstrate the potential for developing new cultivars that have both high yield and are resilient to water stress.

### 3.3. The Physiological Response of Cultivars to Water Stress

Traditionally, phenotypic and/or physiological responses to drought are used to identify tolerance, avoidance or escape strategies in relation to grain yield [24]. Cluster analysis of physiological traits among seven rice cultivars in response to increasing water stress levels generally showed a contrasting response in photosynthesis parameters (A and *g_s_*) vs. the rest of the traits (WUE, SPAD and Leaf Tm). In addition, A and *g_s_* were positively associated with yield (Appendix A and Figure 2). Stomatal closure is one of the earliest and quickest responses for plants to water stress [25]. Moreover, photosynthetic rate is largely dependent on intercellular CO_2_ concentration, whose function is dependent on *g_s_* [26]. Our study also showed that there were cultivar variations in A and *g_s_* responses to different levels of water stress, suggesting different means of adaptation (Appendix A and Figure 2).

To better understand the response of each cultivar to different levels of water stress, the percent change in yield, A and *g_s_* was calculated by comparing the water saturated condition (IRRI_1) to IRRI_2, IRRI_3, and IRRI_4 (Appendix A). For the three high yielding cultivars, PI77, TQNG, and FRCS, when water stress conditions resulted in only a 20% loss in yield, there was a dramatic reduction in *g_s_* (> 50%) but this occurred at different soil water depletion levels (mild -IRRI_2, moderate -IRRI_3, and severe -IRRI_4) for each of the cultivars, respectively (Appendix A). This suggests these cultivars had different levels of tolerance to water stress but responded in the same manner, through rapid stomatal closure to reduce transpiration and sustain yield. Overall, when cultivar performance was compared at each stress level, it was observed that: (1) at the mild stress (IRRI_2) condition, two cultivars, PI77 and LGRU, showed the greatest yield reduction (> 20%) along with the greatest reduction in *g_s_* (25–50%); (2) at the moderate stress (IRRI_3) condition, most cultivars showed more than 40% reduction in *g_s_* except the two cultivars SABR and KBNT that had <15% reduction in *g_s_* as well as A; (3) in the severe stress (IRRI_4) condition, all cultivars showed significant reduction in *g_s_* (> 59%) and A (> 40%), but yield response varied greatly from 6% gain to 77% reduction as compared to IRRI_1. As shown in our study, cultivars vary in tolerance to different stress levels through different physiological and metabolic responses. Selection of breeding lines that have both high yield potential and sustained A and *g_s_* function under water stress will likely result in new cultivars that are resilient to water stress.

A close relationship between SPAD index and chlorophyll content has been reported in both monocot (including rice) and dicot species [27]. Our study showed a general increase in SPAD index in response to increasing stress levels across most cultivars (Appendix A). Several studies reported a decline in SPAD readings in response to drought [28,29,30,31]. However, our study also demonstrated there was a delay in heading with increasing water stress, suggesting a decrease in sink demand, delayed remobilization of carbohydrates to the grain and delayed leaf senescence (Appendix A). We speculate that the difference in SPAD response in our study compared to other reports due to the fact that continuous water stress levels being applied throughout the cropping season versus other studies implementing a specific or intermittent stress treatment.

### 3.4. Cultivar Evaluation: Metabolic Adjustment to Water Stress

It has been reported that accumulation of metabolites is lower in field studies than in other controlled greenhouse studies [32,33,34] due to greater weather related fluctuations that occur under field conditions [35]. However, greenhouse studies have their own confounding effects and, in this study, using field plants grown until grain harvest was considered more relevant to studying response to water stress than in a greenhouse study where the scope of the study (i.e., number of experimental units) may be limited and root growth can be constrained by pot size. In an effort to identify traits that have a major effect on stress tolerance and can be used in cultivar development program, we conducted a season long field experiment using three independent experimental units for each C x I treatment combination as an estimate of error and observed significant differences in physiological and metabolic responses attributed to genotypes and irrigation levels. We investigated intraspecific metabolic adjustment during the critical transition phase (i.e., between vegetative and reproductive stages) as water stress at this phase can alter essential plant physiological functions like photosynthetic capacity that potentially lead to loss in grain yield [8].

Cluster analysis was performed to obtain an unbiased correlation of different metabolites to provide insights into different mechanisms of water stress responses. The Cluster 1 metabolites in our study included fructose, glucose, trehalose, and myo-inositol (Figure 3). Glucose and fructose are well-known contributors to osmotic adjustment in many species [36,37,38]. However, recent study showed glucose triggering stomatal closure in a dose dependent manner in Arabidopsis [39]. Another Arabidopsis study also reported that hexokinase, an enzyme that can phosphorylate fructose and glucose, stimulated stomatal closure in guard cells [40] indicating the role of sugar in stomatal regulation. Traditionally, cool leaf temperature, i.e., high *g_s_* (opened stomata), has long been a criterion for high yield in some species [41,42,43] indicating an importance of stomatal regulation in relation to yield potential. Our study also showed that high yielding cultivars, PI77 and TQNG, had low leaf Tm and high *g_s_* compared to the other, lower yielding, cultivars under non-limiting water condition (Appendix A). We also observed that among cultivars with a similar range of yield potential, stress-triggered stomatal closure was associated with yield potential but not until the water stress condition was severe. For example, among two high yielding cultivars, PI77 showed 51% *g_s_* reduction and 23% yield reduction while TQNG with only 6% *g_s_* reduction produced 19% gain in yield under mild stress (IRRI_2) (Appendix A). For the relationship between sugar accumulation and stomatal regulation under water stress, higher accumulation of Cluster 1 metabolites is not associated with stomatal closure in our study (Appendix A). This implies that a possible role of Cluster 1 metabolites is in osmotic adjustment rather than in stomatal regulation under water stress condition. However, this would need to be verified by measuring leaf water potential in a future study.

Trehalose 6-phosphate (T6P), the intermediate of trehalose biosynthesis, can regulate sugar influx and metabolism [44]. Myo-inositol, a well-known polyol with mannitol, plays a role in scavenging reactive oxygen species (ROS), resulting in prevention of oxidative damage to membranes [37,45,46]. In the current study, accumulation of Cluster 1 metabolites demonstrated differences in cultivar-irrigation responses as seen in Groups 1 and 2 (Figure 3). In contrast, Cluster 4 metabolites, which mainly include amino acids known as stress indicators [37,47], were observed to increase with IRRI_3 and 4 in all cultivars. Branched chain amino acids (Ile, Leu, and Val), plus, Phe, Pro, and Thr have been reported to accumulate in response to stress, and thus are implicated as stress markers. Mannitol, raffinose, and galactinol, known as osmo-protectants, as well as putrescine, are also in Cluster 4. Putrescine is one of the polyamines that play a role in protecting membranes, and thus alleviating oxidative stress [48,49,50].

Previous studies have indicated a role of fructose and glucose in drought tolerance but not specifically to yield response [34]. Interestingly, the cultivars with < 14% yield reduction even under severe water stress (SABR and KBNT), displayed significantly higher accumulation of Cluster 1 carbohydrate metabolites at mild and moderate stress conditions (IRRI_2 and IRRI_3) but no change under severe stress, as compared to 0% water deficit condition. Moreover, the Cluster 1 metabolites group showed a pattern opposite to that of Cluster 4 metabolites group (known as stress responsive metabolites) with increasing water stress (Figure 4). However, Cluster 1 and 4 metabolites were clustered together in the two high yield response cultivars (PI77 and TQNG) (Figure 4) indicating that both soluble sugars and stress indicator metabolites are activated under similar stress condition in these cultivars.

### 3.5. Interrelationship of Physiological and Metabolic Status with Yield Potential and Water Stress

When soil moisture level declines, plants maintain turgor by triggering metabolic functions [51,52,53,54]. Such metabolic functions may include osmotic adjustment to sustain plant performance under water stress conditions but these functions vary between crop species as well as intraspecifically [54,55]. In our study, the cultivars with the greatest yield reduction in response to increasing water stress levels (TQNG and PI77) showed a positive correlation of carbohydrates and stress responsive metabolites (Cluster 1 and 4) with physiological stress responses of leaf Tm, SPAD, and WUE; but were negatively associated with leaf photosynthetic adjustment (i.e., A and *g_s_*) and yield (Figure 5). However, in the low yielding cultivars with little or no yield loss upon stress (KBNT and SABR), the carbohydrate metabolites (Cluster 1, Group 1) response was clearly separated from the stress responsive-like metabolites (Cluster 4, Group 2), with Cluster 1 metabolites positively correlated with leaf photosynthetic adjustment. These results imply that cultivars have different physiological and metabolic responses and those that efficiently regulate the carbohydrate metabolites can minimize yield reduction due to water stress.

## 4. Materials and Methods

### 4.1. Plant Materials And Experimental Conditions

Field studies were conducted on a Dewitt silt loam soil (fine, smectitic, thermic, Typic Albaqualf) at the Dale Bumpers National Rice Research Center/University of Arkansas Rice Research and Extension Center located in Stuttgart, Arkansas during 2014-2016. Each year, the four irrigation treatments were laid out in strips with three field locations of 15 cultivars (3 field locations × 4 irrigation levels × 15 cultivars, 180 experimental units per year) using a randomized complete block (RCB) design within each irrigation strip. The cultivars included commercial varieties and germplasm lines that have been developed or used by southern USA breeders. Each irrigation strip treatment was bordered on each side by two rows planted with a common cultivar and with the same irrigation level as the adjacent treatment to avoid any gradient in the soil moisture content between irrigation treatments. Each experimental unit consisted of two rows planted 41 cm apart and 2.1 m long with the SDI emitter located between the two rows. Experimental units were drill seeded (May 27, 2014; June 12, 2015; and May 4, 2016) using a 118 kg ha^-1^ seeding rate with an Almaco planter (Nevada, Iowa). Fertilizer application and weed control measures were in accordance with local recommendations for rice production. Briefly, a non-selective herbicide was applied early in the spring prior to planting. After planting, Command 3 ME (clomazone) was applied followed by Propanil, Permit plus (halocufuron-methyl), Facet L (dimethylamine salt of quinclorac) and Clincher SF (cyhalofop-butyl) as needed throughout the season at labeled rates for broad spectrum weed control. Approximately three weeks after emergence, 140 kg/ha of nitrogen (46-0-0) was broadcast, followed by irrigation using a portable sprinkler until dissolved and incorporated. Initially, all experimental units were fully irrigated to ensure uniform plant stands and vigorous plant growth. At the V3 stage, experimental units were thinned so that individual plants were on a 31 cm × 41 cm grid allowing for full genotypic yield potential without plant to plant competition. Starting at the V5 stage, an SDI system was used to implement four irrigation treatments of 30% (IRRI_1), 24% (IRRI_2), 20% (IRRI_3), and 14% (IRRI_4) volumetric water content (VWC) for a Dewitt silt loam. These VWCs represent treatments that would correspond to a management allowed depletion levels [56] equivalent to field capacity (or saturated) (FC = 29%, 1% above), 25% deficit, 50% deficit, and just above the wilting point (80% deficit), respectively. Soil texture was 68% silt, 19% clay, and 13% sand; total organic carbon was measured to have 1.01% saturated paste moisture of 35%, with a disturbed bulk density of 1.37 g/cm^3^. The four water treatments were maintained throughout the season (V5 until physiological maturity). For each treatment, 15 mil irrigation tapes (Jain Irrigation, India) were placed at a depth of 20 cm between two planted rows of the experimental units with emitters at 81 cm lateral spacing to ensure uniform application and distribution of water throughout the irrigation zone. A time domain transmissometry moisture sensor (TDT, Acclima, Meridian ID) was placed at the center of each of the irrigation treatment strips and between two emitter lines to monitor soil moisture. According to the soil moisture content measured by the TDT sensors, irrigation was applied automatically by a smart controller (Acclima Inc., Meridian ID) to maintain the target soil moisture content within each irrigation zone (IRRI_1, 2, 3, and 4). Additionally, the irrigation treatment strips were tested every two to three days with a portable soil moisture probe to verify the accuracy of the irrigation controller program which maintained the target soil moisture for each treatment. Following rain events, irrigation did not resume until the soil moisture set points were reached.

### 4.2. Field Measurements

Soil moisture was determined for each irrigation treatment and cultivar per experimental unit every 1 to 2 weeks throughout the growing season using a portable soil moisture sensor (POGO Pro, Stevens Water Monitoring Systems Inc. Portland, OR). Two to four representative plants per experimental unit were selected for data collection throughout the season (e.g., soil moisture, plant height, heading, harvest, physiological traits, metabolic sampling, etc.) and these were the same ones that were harvested for yield. Because the plots were thinned to a uniform stand, the selected plants represented the full genotypic potential of the cultivar at that irrigation level. Data recorded included days to heading and days to harvest maturity from emergence. To calculate grain weight per plant, the representative isolated plants per experimental unit were harvested at maturity, the grain threshed and then dried to 12% moisture. The yield (grain weight per plant) responses of the 15 cultivars to the irrigation treatments in years 2014 and 2015 were used to identify a subset of seven cultivars (Lagrue (LGRU, PI 568891), PI312777 (PI77), Saber (SABR, PI 633624), Francis (FRCS, PI 632447), Kaybonnet (KBNT, PI 583278), Lemont (LMNT, PI 483237), and Teqing (TQNG, PI 536047)), that were selected for metabolic analysis in the 2016 field study. In 2016, at 84 days after emergence (DAE), when the plants were transitioning from the vegetative stage (the 84 DAE sampling period was, on average, 17 days before heading across all cultivars and irrigations), leaf temperature was measured on three upper leaves of two representative plants per experimental unit using a non-contact digital laser infrared thermometer (Etekcity Corporation, Anaheim, CA, USA). At the same time, relative chlorophyll content was measured (SPAD-502 Plus, Konica Minolta Inc., Japan). In addition, measurements of net CO_2_ assimilation (A), stomatal conductance (*g_s_*), and transpiration (E) were determined (Li6800 instrument, LiCOR, Lincoln, NE, USA) using the flag leaves of the two to four representative plants and photosynthetic water use efficiency (A/*g_s_*) was calculated. Saturating red LED light (1800 μmol m^−2^ s^−1^) with 10% blue light of the system was used during the measurements. A CO_2_ cartridge was also used to supply a constant 400 ppm concentration as the reference line setting in the leaf chamber. During gas exchange measurements, the air temperature and the humidity in the leaf chamber were set to match the current environmental conditions, and the vapor pressure deficit (VPD) was set to 1.8 for consistency purposes. All the physiological parameters (A, *g_s_*, Tm, and SPAD) were taken between 11 am and 2 pm and metabolic samplings were taken within 1 to 2 pm. Experimental units within a field replication were measured sequentially so as to minimize any error due to time of day via the field replication effect in the statistical model. Results of the 84 experimental units (3 field locations × 4 irrigation levels × 7 cultivars) with two to four biological replicates in 2016 were analyzed and significant differences determined.

### 4.3. Metabolite Measurements

In the 2016 experiment at 84 DAE, a fully expanded leaf (just below the flag leaf) from three to four representative plants of 84 experimental units (3 field locations × 4 irrigation levels × 7 cultivars), was collected, and immediately frozen with liquid nitrogen for destructive metabolic sampling on the same plants that were ultimately harvested for yield. Approximately 30 mg dry weight (DW) of ground foliar material was extracted from each leaf of three to four biological replicates per experimental unit. The frozen leaf samples were freeze-dried, and then homogenized. These pulverized leaf samples were extracted with 1.4 mL of 70% aqueous methanol containing a mixture of 62.5 nmol of α-aminobutyric acid and 26 μmol of ribitol. Then, 20 μL of each extract was dried and then derivatized with 50 μL of pyridine containing 1 mg of methoxyamine, and incubated for 90 min at 30 °C. Next, 50 μL of MSTFA [N-methyl-N-trimethylsilyl fluoracetamide] was added, incubated for 30 min at 37 °C, and then samples were injected using gas chromatography coupled to mass spectrometry (GC-MS; 7890 GC system, 7693 autosampler, 5975C inert XL MSD; Agilent Technologies, Santa Clara, CA, USA) to measure metabolite content [32,34]. The 40 metabolites measured included Fructose (Frc), Glucose (Glc), myo-inositol (myo-ino), Sucrose (Suc), Ribose (Rib), Mannitol (Mann), Maltose (Mal), Trehalose (Tre), Raffinose (Raff), Galactinol (Gltl), Pinitol (Pin), Succinate (Succ), Citrate (Cit), Fumarate (Fum), Pyruvate (Pyr), Oxalic acid (Oxal), Glycerate (Glyc), Malic acid (Mala), Malonic acid (Malo), Maleic acid (Male), Aconitate (Acon), 2-oxoglutaric acid (2-oxo), Quinic acid (Quin), Shikimic acid (Shik), Adipic acid (Adip), Putrescine (Put), Glycine (Gly), Isoleucine (Ile), Leucine (Leu), Valine (Val), Phenylalanine (Phe), Serine (Ser), Glutamine (Gln), Alanine (Ala), Aspartic acid (Asp), Proline (Pro), Threonine (Thr), Asparagine (Asn), Histidine (His), and Tryptophan (Trp). Standard curves were generated with four dilution series, i.e., 5×, 10×, 20× and 40×, and known mixtures of carbohydrates, organic acids, and amino acids were used for quantifications.

### 4.4. Statistical Comparisons

For hierarchical clustering analysis, the levels of the 29 metabolites, which showed statistical significance of C, I, and/or C x I effects (*p <* 0.01) of harvested samples from IRRI_2 to 4 (24, 20, and 14% VWC; levels of water stress, respectively) were divided by those of IRRI_1 (30% VWC; field capacity), and log2 transformed. The values of photosynthetic rate (A), stomatal conductance (*g_s_*), water use efficiency (WUE; A/*g_s_*), leaf temperature (Tm), and SPAD index were standardized (x-mean/sd) before hierarchical clustering as well as correlation analysis as units and value ranges varied. Statistical analyses, graphical representations of correlation analysis (linear regression analysis), and pairwise comparison (Tukey-Kramer HSD and student *t*-test) were performed using the JMP^®^ software version 12.0.1. Hierarchical clustering heatmap (distance: Pearson correlation) was created using the MultiExperiment Viewer (MeV) program. Principal component analysis (PCA), bivariate fit, and ANOVA were performed using JMP program.

## 5. Conclusions

Adopting water saving management practices in rice production will significantly conserve water resources as well as reduce input (e.g., pumping underground water) costs. However, farmers will not adopt such practices if there is a significant risk of economic loss (i.e., reduced yield and/or grain quality). Having water stress tolerance during the transition from vegetative to the reproductive stage is critical to sustain floret fertility, as water stress during the reproductive stage critically impacts grain filling (i.e., yield and quality). This study, using seven rice cultivars, demonstrated a wide range of responses in physiological and metabolic traits that were related to yield under reduced irrigation. Our study showed that under water stress conditions, cultivars with efficient regulation of soluble sugars, fructose, glucose, and myo-inositol were associated with minimized yield losses, in general. We also found that yield performance of a cultivar was a function of a suite of physiological and metabolic responses to water stress during the transition from the vegetative to the reproductive stage. The results also suggest that there exists cultivar variation in response to different stress levels indicating a genetic balancing between survival (stress tolerance) and grain production. Our study demonstrated that different breeding strategies may be needed to develop new cultivars for deficit irrigation production systems. Under mild stress (mild water saving), high yielding cultivars with a high transpiration rate would likely sustain higher yield than lower yielding with lower stomatal conductance. However, when aggressive water savings is attempted and severe stress conditions can occur, cultivars having lower transpiration rates and also accumulate soluble sugars would be a better option.

## Figures and Tables

**Figure 1 ijms-20-01846-f001:**
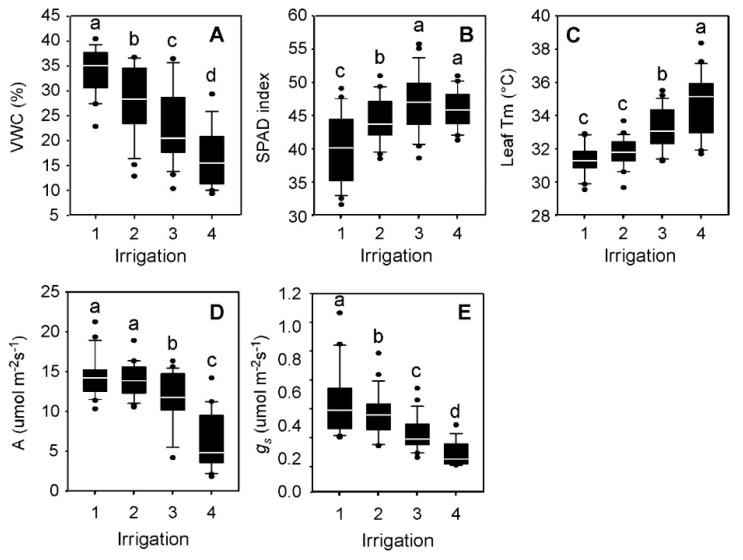
Soil water contents and response of physiological traits to four irrigation treatments in 2016. Soil volumetric water content (VWC; **A**), SPAD index (**B**), leaf temperature (T_m_; **C**), photosynthetic CO_2_ assimilation (A; **D**), stomatal conductance (*g_s_*; **E**) were measured at 84 days after emergence (DAE) just before heading. Irrigation treatments are shown from left to right for IRRI_1 (30% VWC), IRRI_2 (24% VWC), IRRI_3, (20% VWC), and IRRI_4 (14% VWC). The median upper and lower quartiles are indicated as boxes, and dots are means of each of the seven cultivars selected for study. Pairwise comparison was performed using Tukey-Kramer’s test. a, b, c, d letters are statistically different at < 0.05 significance level.

**Figure 2 ijms-20-01846-f002:**
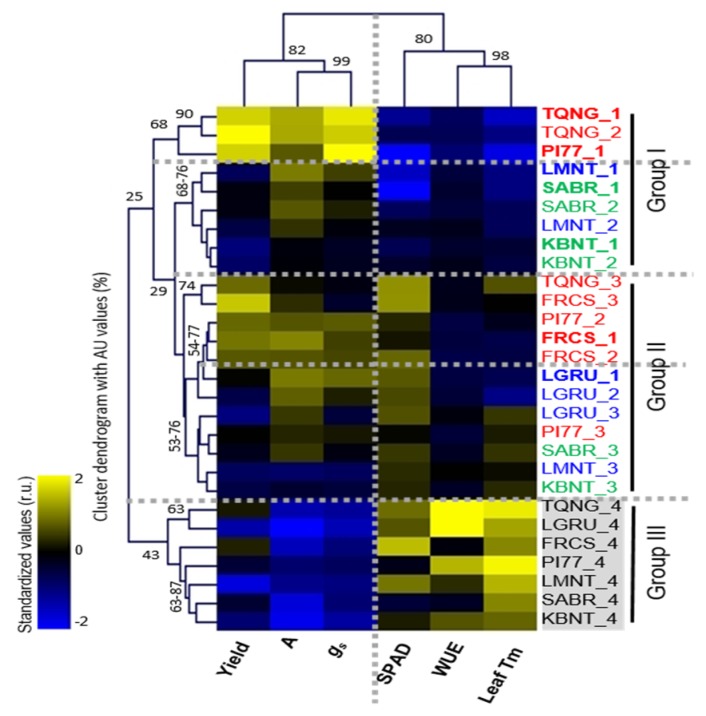
Two-way hierarchical clustering heatmap of relationship among five physiological traits of net CO_2_ assimilation rate (A), stomatal conductance (*g_s_*_)_, water use efficiency (WUE) (A/*g_s_*), relative chlorophyll content (SPAD) index, and leaf Tm as determined by cultivars response to different irrigation levels. Statistical relationships depicting differences between irrigation treatments (IRRI_1 to 4) of physiological traits (horizontal axis) among seven cultivars are shown on the vertical axis (cultivar names are followed by number depicting irrigation regime). Standardized values of physiological trait greater than 0 are in gradations of yellow up to 2× (see scale bar upper left), ratios that were unchanged are shown in black and values less than 0 are in gradations of blue. AU indicates approximately unbiased *p*-values (0–100%, with higher numbers denoting greater significance).

**Figure 3 ijms-20-01846-f003:**
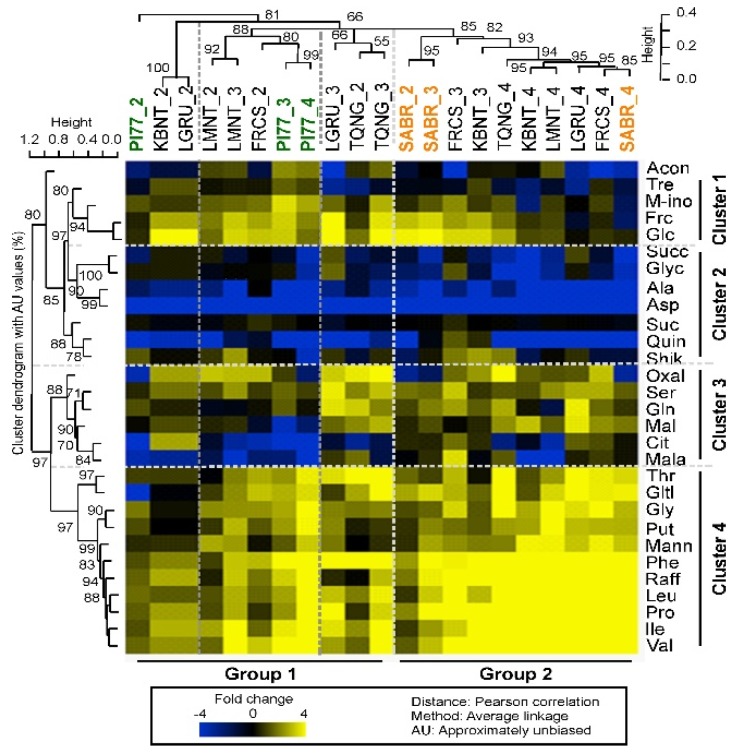
Two-way hierarchical clustering heatmap of metabolic changes in response to different irrigation levels of seven rice cultivars. Metabolite ratios greater than 0 are in gradations of yellow up to 4-fold increase compared to IRRI_1 condition (see scale bar), ratios that are unchanged are shown in black and ratios less than 0 are in gradations of blue up to 4-fold decrease. Distance was determined by Pearson correlation coefficient. AU indicates approximately unbiased *p*-values (0–100%, with higher numbers denoting greater significance). Abbreviations of cultivars are for TQNG, Teqing; PI77, PI312777; LGRU, Lagrue; LMNT, Lemont; FRCS, Francis; SABR, Saber; KBNT, Kaybonnet. Cultivar names are followed by number depicting irrigation regime. Abbreviations of metabolites are for Acon, aconitate; Frc, fructose; Glc, glucose; myo-ino, myo-inositol; Tre, trehalose; Ala, alanine; Asp, aspartate; Succ, succinate; Glyc, glycerate; Quin, quinate; Shik, shikimate; Oxal, oxalate; Ser, serine; Gln, glutamine; Mal, maltose; Cit, citrate; Mala, malate; The, threonine; Gltl, galactinol; Phe, phenylalanine; Ile, isoleucine; Val, valine; Leu, leucine; Pro, proline; Raff, raffinose; Mann, mannitol; Gly, glycine; Put, putrescine.

**Figure 4 ijms-20-01846-f004:**
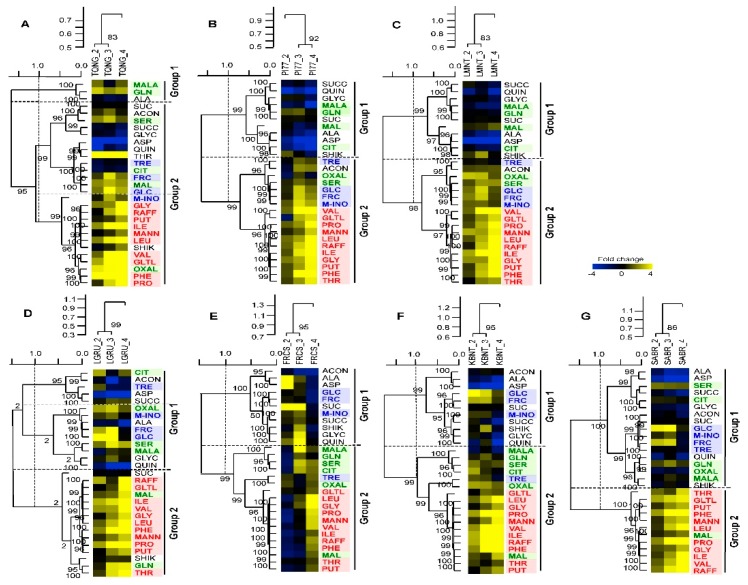
Two-way hierarchical clustering heatmaps of metabolic changes in response to different irrigation levels of seven cultivars from Figure 3. Cultivars are classified as high (TQNG (**A**), PI77 (**B**)), intermediate (LMNT (**C**), LGRU (**D**)) and low (FRCS (**E**), KBNT (**F**), SABR (**G**)) yield response as in Table 1. Metabolites in blue, green, and red are Cluster 1, 3, and 4, respectively, as designated in Figure 3. Other details are as in Figure 3.

**Figure 5 ijms-20-01846-f005:**
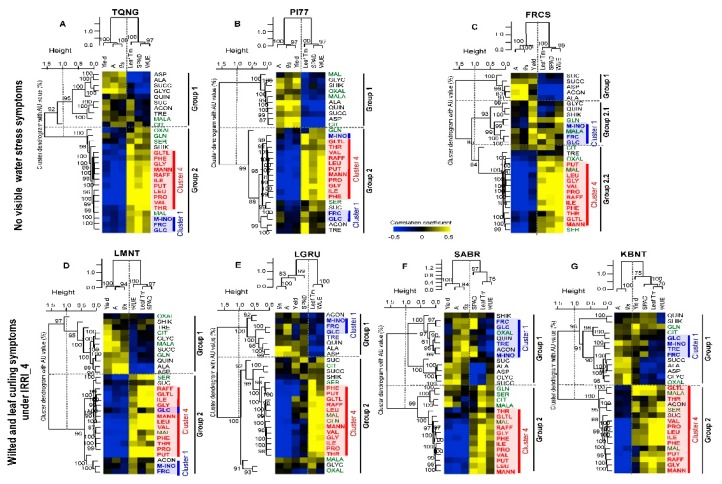
Two-way hierarchical clustering heatmaps of metabolic changes in relation to physiological traits and yield. TQNG (**A**), PI77 (**B**), and FRCS (**C**) have no visible water stress symptoms while LMNT (**D**), LGRU (**E**), SABR (**F**), and KBNT (**G**) displayed wilted and leaf curling symptoms at IRRI_4. Values are correlation coefficients between each physiological trait (i.e., A, *g_s_*, relative chlorophyll content, leaf Tm, WUE (A/*g_s_*), and yield with metabolite contents. Metabolites in blue, green, and red are Cluster 1, 3, and 4, respectively as designated in Figure 3. Other details are as in Figure 3.

**Figure 6 ijms-20-01846-f006:**
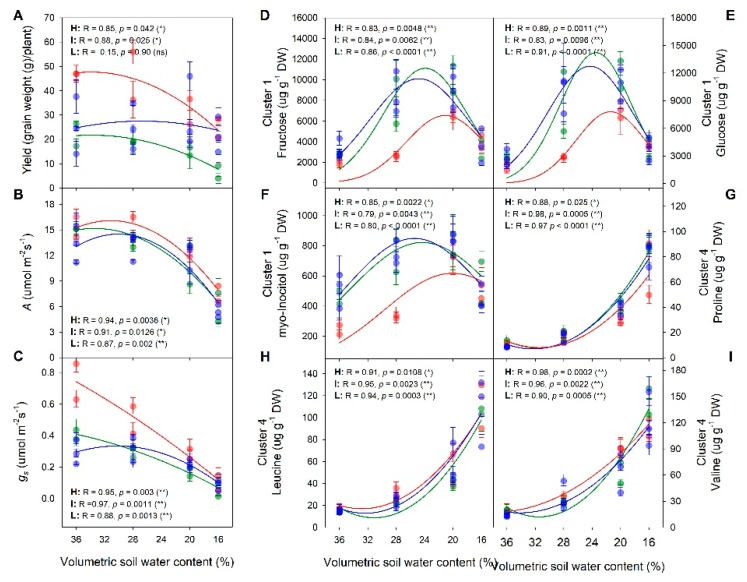
Trait responses as a function of volumetric soil water content (%) of four irrigation regimes among high, intermediate, and low response cultivar groups. Bivariate fit of yield (**A**), assimilation rate (**B**), stomatal conductance (**C**), fructose (**D**), glucose (**E**), myo-inositol (**F**), proline (**G**), leucine (**H**), valine (**I**) in response to increasing soil water stress levels are presented. Two high, two intermediate, and three low response cultivars are listed as H (red), I (green), and L (blue), respectively. Dots with error bars are avr ± sem, lines are regression plots, and **, * and ns indicate *p* < 0.01, *p* < 0.05 and *p* > 0.05, respectively.

**Table 1 ijms-20-01846-t001:** Means and regression analysis of grain weight per plant in response to four irrigation levels for each cultivar in 2014, 2015, and 2016. **, *, *ns* indicate *p <* 0.01, *p <* 0.05, *p >* 0.05, respectively.

Cultivar	Year	Yield under Water Saturated Condition (IRRI_1)		Yield Response (Slope of yield/% VWC)				Yield Performance under Water Stress
		Mean	SE	Estimate	SE	Prob. > F	R^2^	
**TQNG**	**2014**	38.77	4.97	2.24	0.45	**	0.72	High response
	**2015**	55.80	8.05	2.37	0.82	*	0.55	
	**2016**	47.00	2.78	1.95	0.64	**	0.54	
**PI77**	**2014**	31.60	6.13	1.59	0.46	**	0.55	
	**2015**	37.30	8.99	1.00	0.52	ns	0.35	
	**2016**	47.00	3.51	1.57	0.45	**	0.56	
**LGRU**	**2014**	25.00	1.67	1.02	0.25	**	0.62	Intermediate response
	**2015**	13.60	0.40	0.09	0.32	ns	0.02	
	**2016**	26.50	1.04	1.33	0.33	**	0.34	
**LMNT**	**2014**	16.93	0.73	0.80	0.27	**	0.47	
	**2015**	15.43	3.91	0.10	0.58	ns	0.01	
	**2016**	17.33	2.77	0.90	0.29	**	0.62	
**FRCS**	**2014**	38.03	1.68	0.89	0.72	ns	0.13	Non or low response
	**2015**	31.13	5.40	0.41	0.58	ns	0.08	
	**2016**	37.67	6.89	0.33	0.49	ns	0.04	
**KBNT**	**2014**	16.03	6.53	0.56	0.55	ns	0.09	
	**2015**	13.10	3.41	0.49	0.20	ns	0.50	
	**2016**	14.17	5.13	−0.10	0.32	ns	0.01	
**SABR**	**2014**	30.63	3.54	0.70	0.38	ns	0.25	
	**2015**	19.00	3.06	0.09	0.24	ns	0.02	
	**2016**	24.50	0.76	0.21	0.22	ns	0.08	

**Table 2 ijms-20-01846-t002:** Relative yield reduction of seven rice cultivars under four irrigation levels (IRRI_1 to 4) in 2016. R section represents three years’ response based on regression. H, high responsive cultivar; I, intermediate responsive cultivar; and L, low responsive cultivar.

Cultivar	Based on 2014-2016 Results									2016 Results		
	R	Mean Yield (g)					% Yield Loss from IRRI 1			% Yield Loss from IRRI 1		
		IRRI1	IRRI2	IRRI3	IRRI4	Overall	IRRI2	IRRI3	IRRI4	IRRI2	IRRI3	IRRI4
**TQNG**	**H**	47.19	51.51	31.55	17.56	36.95	−9.16	38.75	62.80	−19.15	21.99	39.36
**PI77**		38.63	26.09	21.62	15.08	25.35	32.47	17.14	60.97	22.70	43.97	56.03
**LGRU**	**I**	21.70	19.41	16.20	10.09	16.85	10.55	16.54	53.48	28.93	49.06	64.78
**LMNT**		16.57	14.02	13.40	7.40	12.85	15.36	4.44	55.33	−10.58	2.88	76.73
**FRCS**	**L**	35.61	33.48	36.50	25.58	32.79	5.99	−9.03	28.17	7.52	−22.12	22.12
**KBNT**		14.43	11.95	18.90	8.23	13.38	17.21	−58.16	42.96	−14.12	−36.47	−5.88
**SABR**		24.71	26.66	23.27	20.18	23.70	−7.87	12.71	18.35	0.68	4.76	13.61

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
