# Peer review of "Physiological and Metabolic Responses of Rice to Reduced Soil Moisture: Relationship of Water Stress Tolerance and Grain Production"

_ijms, 2019, doi:10.3390/ijms20081846_

Round 1

Reviewer 1 Report

The article describes a three years prepared field experiment, run out in one season, aimed at understanding the different physiological behaviour of different cultivars of rice subjected to four watering regimes. The authors collected a huge number of different traits: yield; photosynthetic traits and metabolites.

The article is well written and the overall experiment is well designed, with an enormous set of data and with a methodology, in general, correctly handled. However,I think that the statistical treatment of data based on a two-way hierarchical clustering, drives to a description of results more than a mechanistic interpretation of physiology of response to water stress in rice. Furthermore, it miss a main clear hypotesis. By consequence, the discussion suffers of it. I suppose that using a generalized linear model authors might find the best model to find out which traits could mainly drive the output variables such as yield and photosynthetic assimilation. I am convinced that this important set of data can lead to new interesting insights in rice drought response.

Other general concerns and suggestion

I don’t understand why the authors always write “physiological and metabolic ….” as if the two things are different. I would remember that the cellular metabolism is a part of whole plant physiology!

I have some doubts that extracting the cellular components with a single method can be adapt to such a different pattern of metabolites.

I can understand that the methodology utilized for metabolites cannot be used for starch analysis, but I think that the amount of this carbohydrate is quite important to the right interpretation of the role of soluble sugars during water stress in rice.

The authors argue that one of the main conclusion that can arise from the interpretation of data is that under water stress conditions, cultivars with efficient regulation of carbohydrates, having a role in osmotic adjustment, were able to minimize yield losses. To really say that, I think that among this huge set of data it was missing the measure of hydraulics for example the water leaf potential.

Minor concerns

Please see the attached documents

Author Response

Our response is attached here. Thank you.

Reviewer 2 Report

Review of IJMS 464130. Physiological and metabolic responses of rice to reduced soil moisture: relationship of water stress tolerance and grain production. By Jinyoung Y. Barnaby et al.

In my view this manuscript provides findings on three main fronts: firstly that the resilience of yield response to limiting water availability shows genetic variation among rice genotypes,  secondly, that this is not correlated with yield under well –watered conditions, and thirdly, it may be possible to use metabolomics profiles, and certain specific groups of metabolites as biomarkers to follow this trait. These findings are of significance due to the decreasing water availability occurring and predicted to occur in future in rice growing areas and the necessity to use limit-irrigation by some means without decreasing yield.

I found the paper describes a very detailed body of work, generally well written, and the conclusions are supported by the results, as far as I can judge. Besides several minor corrections/revisions listed below, I have two other comments regarding the presentation for the authors to consider:

In the results, I found the presentation very hard going with regard to the two-way hierarchical clustering descriptions (Figs 2-5). While I accept that this may be an appropriate way to deal with a large body of data, (and I admit I am not especially familiar with this approach), I suggest some brief detail of the method at the start of this approach may be helpful to many readers. For example it was not obviously clear to me what data is represented on the y axis in Fig 2. In addition, these results represent a large amount of data per figure, and relatively long descriptions, and I wonder if some parts could be shortened or abbreviated without taking away from the message.

Secondly, I find that the overall conclusions are somewhat tentative –that it will be necessary to ‘combine physiological and metabolic response mechanisms’ to develop new cultivars with water stress tolerance. Surely, from the findings of this substantial work, the authors are able to make more specific and more firm recommendations of the kinds of changes that should be sought.

Minor revisions/queries

L 447 and Abbreviations: complete block design ?

L 472 mil, mm ?

L 507, to obtain 400 ppm CO2, does this represent the reference or sample line setting ?

L517-518 How were leaves stored, dried after collection, and pulverised before extraction ?  Extraction temperature and time ?

L 539 Normalised using what ?

Author Response

(The authors gave the same response as above.)

Reviewer 3 Report

The manuscript entitled" Physiological and metabolic responses of rice to reduced soil moisture: relationship of water stress tolerance and grain production"  The object is very clear and the results will have  good potential for biotechnology and agriculture.  The manuscript also provide systemic and scientific strategies to assess the adequate irrigation resources in rice at the critical transition between vegetative and reproductive growth stages. 

However, the authors claim that the existence of genetic variation in yield under different water stress levels which results from a suite of physiological and biochemical responses to water stress, and not from a single dominant trait affecting final yield. The statement should be carefully demonstrated experimentally. It was reported that genetic modification of one transcription factor can improve tolerance to water stress in rice. Please refer to the following references as an example.

https://academic.oup.com/jxb/article/68/16/4695/4082069

https://www.frontiersin.org/articles/10.3389/fpls.2018.00094/full

https://www.ncbi.nlm.nih.gov/pmc/articles/PMC5299611/

https://journals.plos.org/plosone/article?id=10.1371/journal.pone.0030765

https://www.ncbi.nlm.nih.gov/pmc/articles/PMC3661581/

Thus, it is suggested that the authors should analyze the some gene biomarkers of all of the cultivars. Alternatively, inclusion of some transgenic rice with tolerance to water stress in the experiments would be helpful to clarify if a single dominant trait affecting final yield. The discussion of genetic-modified/transgenic rice with tolerance to water stress should be included. 

Author Response

Reviewer 3.

The manuscript entitled" Physiological and metabolic responses of rice to reduced soil moisture: relationship of water stress tolerance and grain production" The object is very clear and the results will have good potential for biotechnology and agriculture. The manuscript also provides systemic and scientific strategies to assess the adequate irrigation resources in rice at the critical transition between vegetative and reproductive growth stages. 

However, the authors claim that the existence of genetic variation in yield under different water stress levels which results from a suite of physiological and biochemical responses to water stress, and not from a single dominant trait affecting final yield. The statement should be carefully demonstrated experimentally. It was reported that genetic modification of one transcription factor can improve tolerance to water stress in rice. Please refer to the following references as an example.

https://academic.oup.com/jxb/article/68/16/4695/4082069

https://www.frontiersin.org/articles/10.3389/fpls.2018.00094/full

https://www.ncbi.nlm.nih.gov/pmc/articles/PMC5299611/

https://journals.plos.org/plosone/article?id=10.1371/journal.pone.0030765

https://www.ncbi.nlm.nih.gov/pmc/articles/PMC3661581/

Thus, it is suggested that the authors should analyze some gene biomarkers of all of the cultivars. Alternatively, inclusion of some transgenic rice with tolerance to water stress in the experiments would be helpful to clarify if a single dominant trait affecting final yield. The discussion of genetic-modified/transgenic rice with tolerance to water stress should be included.

Thank you for your suggestion. We understand that mutants and genetically modified lines can be a way to determine the impact of a single gene/trait on tolerance to drought stress. However, the US rice industry does not allow transgenic materials in commercial rice production and there are severe restrictions/prohibitions on conducting field research using transgenic materials, even for proof of concept. Thus, this is not a viable option for us. Moreover, the mandate of our research program is to fully explore naturally existing genetic variation for the development of improved rice varieties and understanding trait-gene function. We, therefore, examined 15 cultivars, and selected seven displaying varied yield response to water stress for our study. Our purpose of study was to identify metabolic or photosynthetic traits that can be used in a conventional (non-transgenic) breeding program. We have now added a section to describe the complex inheritance of drought tolerance and the role of transgenic technology in these studies using additional and your suggested references (Lines 77-88). Thank you.

Round 2

Reviewer 3 Report

The authors have responded to my concerns. Editing of English is required.